# Genetic Gains and Field Validation of Synthetic Populations in Tropical Maize Using Selection Indexes and REML/BLUP

**DOI:** 10.3390/plants14203149

**Published:** 2025-10-13

**Authors:** Antônia Maria de Cássia Batista de Sousa, Marcela Pedroso Mendes Resende, Ailton Jose Crispim-Filho, Glauco Vieira Miranda, Edésio Fialho dos Reis

**Affiliations:** 1Escola de Agronomia, Universidade Federal de Goiás, Goiânia 74690-900, Brazil; antonia.sousaufpi@gmail.com (A.M.d.C.B.d.S.); marcelapmr@ufg.br (M.P.M.R.); ailton.crispimfilho@gmail.com (A.J.C.-F.); 2Agronomy Coordination, Federal Technological University of Paraná, Santa Helena 85892-000, Brazil; glaucovmiranda@utfpr.edu.br; 3Instituto de Biociências, Universidade Federal de Jataí, Jataí 75801-615, Brazil

**Keywords:** *Zea mays*, inbreeding depression, heterosis

## Abstract

The development of tropical maize populations with high heterosis potential is essential for sustaining genetic progress in hybrid breeding programs, yet accurate selection remains challenging due to genotype–phenotype interactions and inbreeding depression. This study evaluated the efficiency of five selection strategies in recurrent selection programs using F_2_ populations derived from commercial maize hybrids: Smith–Hazel Index (SHI), Base Index (BIA), Mulamba–Mock Index (MMI), REML/BLUP for grain yield (BLUP_GY), and REML/BLUP for inbreeding depression (BLUP_ID). Consistency among methods was assessed with a heatmap, and predicted genetic gains were compared with realized field performance. Predicted gains were highest with MMI and BIA for grain yield and ear weight, although realized results revealed discrepancies, particularly for BLUP-based approaches. Notably, BLUP_GY, which had the lowest predicted yield (4025 kg ha^−1^), achieved a realized yield of 5620 kg ha^−1^, surpassing BIA and SHI. This indicates that additive potential was underestimated in predictions, likely due to dominance and environmental effects in early F_2_ cycles. Overall, BLUP-based methods proved effective in identifying progenies with higher additive value, and their integration with phenotypic indices is recommended to combine short-term realized gains with sustained genetic improvement.

## 1. Introduction

The development of new tropical maize (*Zea mays* L.) populations with high heterosis potential is a central objective in hybrid breeding programs, especially in the face of climate change and the rising global demand for food. Continuous genetic progress is essential for these programs and relies on efficient strategies to accurately identify superior progenies and form high-performing heterotic groups. However, selection in early recurrent cycles remains challenging due to the complex interaction between genotypic and phenotypic components, further complicated by inbreeding depression.

Traditional phenotypic selection methods, such as mass selection or classical selection indexes, have historically played an important role in breeding pipelines. Nevertheless, their effectiveness is often constrained by environmental variability and their limited ability to account for genetic relationships and additive variance [1]. In this context, modern statistical approaches—especially mixed models based on Restricted Maximum Likelihood (REML) and Best Linear Unbiased Prediction (BLUP)—have become increasingly prominent due to their capacity to generate accurate genotypic estimates under unbalanced designs and across multi-environment trials [2,3].

Several studies have highlighted the advantages of using REML/BLUP in tropical maize breeding. The mixed models could explain more than 99% of phenotypic variance in key traits while maintaining stability under genotype-by-environment interactions [3]. The predictive performance is retained even with up to 20% of hybrids or 23% of environments missing, attesting to the robustness of the methodology [2]. Pedigree-based BLUP led to superior genetic gains in popcorn expansion volume (1–45%), outperforming mass selection [4].

The potential of REML/BLUP in driving genetic gains in tropical maize has been confirmed in recent studies, with improvements of up to 24.07% in traits such as ear length and grain weight in half-sib progenies [5]. Similarly, annual gains between 46 and 118 kg ha^−1^ year^−1^ in early-maturing maize hybrids developed by CMMIYT across 68 environments in seven African countries have been documented, highlighting the effectiveness of structured breeding programs and robust predictive tools in diverse and stress-prone conditions [6].

REML/BLUP outperformed classical selection indexes in predicting genetic gains and estimating genotypic values in supersweet corn programs [7]. While both REML/BLUP and the Mulamba and Mock index showed high coincidence under balanced conditions, REML/BLUP demonstrated superior gains and reliability in multi-trait selection. However, in low-resource settings or early recurrent selection cycles with limited data, simpler phenotypic selection methods remain relevant, with the Mulamba and Mock index standing out for its simplicity, flexibility, and consistent performance across crops such as maize and sugarcane [1,8].

Advances in quantitative genetics, statistical modeling, and molecular tools have significantly improved maize breeding efficiency. The integration of genomic selection (GS) with strategies tailored to trait heritability and genetic architecture has been shown to accelerate genetic gains while preserving genetic diversity [9]. Reciprocal recurrent genomic selection (RRGS) has emerged as a powerful approach to improve both general and specific combining abilities. Comparative analyses between full-sib and half-sib training sets reveal that full-sib strategies often result in higher cumulative selection gains, particularly when specific combining ability (SCA) plays a prominent role [10,11].

Genetic diversity remains a cornerstone of successful breeding programs. Studies using SSR markers in convergent-derived maize populations and in introgressed progenies from wild relatives like *Zea nicaraguensis* emphasize the importance of broadening the genetic base to enhance selection response and population resilience [12,13]. In Brazil, substantial genetic gains have been achieved through intrapopulation recurrent selection, with high heritability and consistent improvement across five selection cycles in fresh corn half-sib progenies, and confirmation of the genetic potential of southern Brazilian maize populations even under low-input conditions [14,15].

Reciprocal recurrent selection with full-sib progenies has also been effectively applied to enhance both heterosis and parental line development, consistent with theoretical models highlighting the benefits of optimizing training sets, maintaining genetic variability, and balancing short- and long-term selection responses [10,16,17]. Thus, integrating REML/BLUP with classical selection indexes—and potentially with ensemble learning or deep learning models—represents a promising strategy for modern breeding optimization [18].

This study builds upon these advancements by comparing the efficiency of three classical selection indexes (Smith–Hazel, Base Index, and Mulamba–Mock) and the REML/BLUP methodology for grain yield and inbreeding depression in selecting 194 half-sib progenies and their S1 progenies, focusing on grain yield and inbreeding depression. By evaluating predicted and realized genetic gains after recombination, the objective is to identify the most effective strategies for constructing genetically diverse base populations with strong heterotic potential for tropical maize. Ultimately, this research addresses the need for accurate, scalable, and efficient selection methods to support the rational formation of high-performing synthetic populations adapted to tropical environments. In doing so, it contributes to sustainable hybrid maize breeding through the integration of quantitative genetics, advanced statistical modeling, and field-based experimentation.

## 2. Results

### 2.1. Selected Progenies for Grain Yield and Inbreeding Depression

The heatmap illustrates the overlap of selected maize progenies among five different selection methods, BIA, MMI, SHI, BLUP_GY, and BLUP_ID, revealing meaningful patterns of convergence and divergence that provide insights into the consistency and complementarity of phenotypic and genotypic approaches in early recurrent selection cycles (Figure 1). A consistent pattern was observed for progenies 22, 39, 43, 135, and 152, which were selected simultaneously by all methods, indicating strong agreement among the different approaches. Progenies such as 101, 113, and 137 were selected by four methods, reflecting high but not complete convergence. In contrast, progenies like 145, 19, 17, and 177 were identified by only two methods, demonstrating limited consensus. A considerable number of progenies were exclusively selected by BLUP_GY and BLUP_ID, without overlap with the index-based strategies (SHI, BIA, and MMI). Additionally, some progenies, including 35, 89, 90, 131, 134, 147, and 168, appeared in only one method, revealing unique selections not shared across approaches. Overall, the results reveal the presence of both widely recurrent progenies across all selection strategies and method-specific progenies, evidencing distinct patterns of consistency and divergence among the applied methodologies.

The coincidence index heatmap shows the degree of overlap in the top 10 selected maize progenies among five different selection methods (Figure 2). The results show a strong agreement between the genomic selection strategies BLUP_GY and BLUP_ID (coincidence index = 0.82), suggesting consistency in the ranking of top-performing progenies based on genomic predictions. In contrast, classical phenotypic index-based methods such as SHI, BIA, and MMI exhibited moderate to low overlap among themselves (e.g., BIA vs. MMI = 0.67; SHI vs. BIA = 0.33) and no overlap with BLUP-based approaches (coincidence index = 0.00 for all comparisons between BLUP_GY or BLUP_ID with SHI, BIA, or MMI).

The heatmap using the top 20 progenies prioritized by each method presents the coincidence index calculated between five selection strategies (Figure 3). The genomic methods BLUP_GY and BLUP_ID exhibited a high concordance (coincidence index = 0.82), confirming strong consistency between their selection outputs. In contrast, phenotypic index-based strategies (SHI, BIA, MMI) showed moderate agreement among themselves—especially BIA vs. MMI (0.74) and SHI vs. MMI (0.48)—while exhibiting minimal overlap with the genomic strategies (only 0.05 in all comparisons between BLUP_GY or BLUP_ID and SHI, BIA, or MMI).

To evaluate the convergence trend among strategies, we calculated the average pairwise coincidence index as a function of the number of top-ranked progenies. The average index increased steadily from 0.23 (top 10) to 0.39 (top 40).

### 2.2. Predicted Selection Gains Based on Phenotype Indexes

The greatest predicted genetic gains with 20% of selected progenies were observed for grain yield (GY) and ear weight (EW), especially using the MMI and BI indexes (Table 1). The MMI index predicted the gain for grain yield (15.93%), followed closely by BI (15.84%) and SHI (14.79%). For ear weight, gains were also substantial, with values ranging from 13.41% (SHI) to 14.01% (BI). Inbreeding depression (ID), considered a key trait in this analysis, exhibited strong negative values across all indexes (SHI: −14.45%, BI: −14.96%, MMI: −15.07%).

Moderate selection gains were recorded for traits such as number of ears per plant (NE), ear length (EL), and ear diameter (ED), with NE showing the most consistent response across methods—SHI: 7.56%, BI: 6.19%, and MMI: 6.11%, confirming its capacity to balance multiple traits when no single trait dominates the selection objective.

In contrast, flowering traits (female and male flowering, FF and MF), plant height (PH), and ear height (EH) exhibited negligible gains across all indexes, indicating limited selection pressure or unfavorable correlations with the main selection target (grain yield) suggesting a favorable trend toward more compact plants, which is desirable in tropical environments for lodging resistance and mechanical harvesting. For example, female flowering (FF) showed negative gains in all cases, ranging from −0.05% (SHI) to −0.19% (BI), suggesting a potential delay in flowering as a correlated response, which may contribute to reduced crop cycle and better adaptation to the second season.

### 2.3. Prediction of Genotypic Values

The evaluation of grain yield in 194 half-sib maize progenies from the third progeny generation (3G) using REML/BLUP methodology revealed substantial variability in both the predicted additive genotypic values (μ_i_ + g_i_) and associated inbreeding depression (u_i_ + g_i_) (Table 2).

Among the 38 selected progenies, progeny 135 ranked at the top by point estimates for predicted grain yield (μ_i_ + g_i_ = 4890 kg ha^−1^) and BLUP (1423 kg ha^−1^), showed low inbreeding depression (33.7), and was selected by all methods, with no statistically significant differences versus other top entries (Table 2; Figure 1).

Notably, the high predicted additive values (μ_i_ + g_i_) and low inbreeding depression observed in selected progenies, such as 135 and 145, demonstrate that modern breeding methods can recover and enhance yield potential without compromising genetic stability.

Similarly, progeny 145 demonstrated a predicted yield of 4525 kg ha^−1^ and a modest inbreeding depression (35.2), reinforcing its potential for selection. Other high-performing progenies, such as 85 (4486 kg ha^−1^) and 150 (4212 kg ha^−1^), also ranked among the best in terms of grain yield; however, their inbreeding depression values were relatively higher (40.10 and 39.8, respectively).

Conversely, some progenies exhibited consistently lower performance, such as 26, 32, 43, and 7 recorded BLUPs below 410 kg ha^−1^ and genotypic values (μ_i_ + g_i_) under 3900 kg ha^−1^, suggesting limited contribution to yield improvement. These genotypes may be excluded from further breeding unless they display desirable traits for other agronomic purposes.

The correlation between BLUPs and predicted additive values was positive, indicating that higher phenotypic expression is generally associated with greater additive genetic merit. This is reflected in the clustering of selected progenies above the average thresholds for both parameters (BLUP = 558 kg ha^−1^; μ_i_ + g_i_ = 4025 kg ha^−1^).

Furthermore, the evaluation across three breeding generations (1st, 2nd, and 3rd) reinforces the heritability and stability of grain yield traits. Genotypes such as 135, 145, and 85, which maintained superior performance across generations, are particularly valuable for integration into long-term breeding strategies.

The average predicted yield for the 38 selected progenies was 4025 kg ha^−1^, with a corresponding average inbreeding depression of 43.1, and a predicted genetic gain of 558 kg ha^−1^ for grain yield and a reduction of 7.90 units for inbreeding depression. The simultaneous consideration of both parameters, high genotypic values for yield and low inbreeding depression, allowed the identification of nine recurrent progeny (135, 145, 85, 150, 87, 4, 44, 42, 96 and 105) that showed consistent performance across generations. Notably, these progenies originated from first-generation progenies previously selected for resistance to foliar diseases, such as gray leaf spot. This convergence highlights the possibility that such progenies carry favorable alleles for both yield and stress tolerance, enhancing their breeding value and justifying their recombination in population improvement programs.

In summary, the results confirm the genetic superiority of a core set of progenies within RV-02, especially those descending from progeny 23, reinforcing the potential of this subgroup as the foundation for developing high-performing, genetically stable tropical maize populations.

### 2.4. Synthetic Populations Performance

The comparison of means among five maize populations developed via different selection strategies, along with two commercial checks (hybrid P3898 and open-pollinated variety AL Bandeirante), revealed significant genetic progress for grain yield (GY) and other yield-related traits (Table 3).

Among the synthetic populations, PopMMI and PopBLUP_GY demonstrated the highest agronomic performance, exhibiting statistically superior ear weight and grain yield, significantly outperforming the open-pollinated check (AL Bandeirante, GY = 4.48 t ha^−1^). This yield surpassed that of the base populations (PopSHI, PopBIA, and PopBLUP_ID) and remained statistically inferior to the commercial check P3898 (9.40 t ha^−1^). Although not statistically equal to P3898, the BLUP_GY population showed evidence of additive genetic gain, validating the use of mixed-model approaches in early-stage recurrent selection cycles. On the other hand, PopBIA, PopSHI, and PopBLUP_ID exhibited lower grain yield and ear weight values, ranging from 4.65 to 4.81 t ha^−1^ for GY and 5.90 to 6.10 t ha^−1^ for EW. Despite being inferior to PopMMI and PopBLUP_GY, these populations still performed comparably or better than the open-pollinated check, indicating partial gain through selection. However, further cycles of recurrent selection or incorporation of genomic tools may be required to boost their yield potential.

Male flowering (MF) did not differ significantly among most populations, except for P3898, which was slightly later (67.83 days) than the experimental groups (65.33 to 66.67 days). This suggests that selection did not strongly impact flowering time, allowing gains in yield without major shifts in phenology.

A comparison between predicted grain yield (Predict GY) and the realized values of grain yield (GY) revealed important discrepancies among the populations evaluated. Among the synthetic populations derived from selection indices, the predicted genetic gains were consistently higher than the realized values. For example, PopMMI presented a predicted GY of 4590 kg ha^−1^, whereas the realized value was 6300 kg ha^−1^, indicating that the prediction underestimated the actual performance. A similar pattern occurred with PopBIA (predicted 4597 vs. realized 4760 kg ha^−1^) and PopSHI (predicted 4515 vs. realized 4650 kg ha^−1^). In these cases, although the absolute difference was modest, the realized values slightly exceeded the predictions, suggesting that the additive genetic effects captured by the indices were expressed more favorably under the experimental conditions.

Conversely, for PopBLUP_GY and PopBLUP_ID, the realized yields were 5620 and 4810 kg ha^−1^, respectively, while the predictions were 4025 and not available for BLUP_ID. Here, the differences were larger, particularly in PopBLUP_GY, where realized GY was substantially higher than predicted. This divergence may be attributed to environmental contributions or non-additive effects (dominance, epistasis) not fully accounted for by the BLUP-based predictions.

The comparison between predicted (Predict GY) and realized GY values highlights consistent differences across the evaluated populations. In general, the realized yields exceeded the predicted values, although the magnitude of the difference varied depending on the selection strategy. For instance, PopMMI showed a realized GY of 6300 kg ha^−1^, compared with a predicted yield of 4590 kg ha^−1^, while PopBIA and PopSHI also exhibited higher realized values (4760 and 4650 kg ha^−1^) relative to their respective predictions (4597 and 4515 kg ha^−1^).

In contrast, the populations based on BLUP methodologies presented distinct outcomes. PopBLUP_GY had a predicted yield of 4025 kg ha^−1^, whereas the realized yield reached 5620 kg ha^−1^, a substantial difference of more than 1500 kg ha^−1^, indicating that additive effects captured in the prediction were sufficient to fully explain the realized performance under field conditions. PopBLUP_ID, on the other hand, did not include a predicted yield in the dataset, but the realized value was 4810 kg ha^−1^, positioning it slightly below PopBLUP_GY in terms of actual performance.

## 3. Discussion

The overlap of selected progenies among the five strategies provides important insights into the consistency and divergence of phenotypic and BLUP-based approaches in early recurrent selection. The recurrent identification of progenies 22, 39, 43, 135, and 152 across all methods suggests that these progenies combine both favorable phenotypic performance and stable predicted genetic merit, reinforcing their potential as reliable candidates for advancement in breeding pipelines.

Overall, the combination of consistent and method-specific selections emphasizes the complementarity of phenotypic indices and BLUP-based approaches, reinforcing the importance of integrating multiple strategies to increase the robustness of selection decisions. This integration can maximize genetic gain by balancing convergence on superior and stable progeny with the exploration of unique candidates that expand the genetic base for long-term breeding progress.

In line with previous studies, the MMI showed strong performance in concentrating desirable traits and suppressing undesirable ones [1,19]. The consistent reductions in flowering time, plant height, and inbreeding depression, alongside improvements in yield components, indicate that this index is highly suitable for simultaneous multi-trait selection in maize half-sib populations under tropical conditions.

The average pairwise coincidence index as a function of the number of top-ranked progenies (top 10 to top 40) indicates that as more progenies are included in the ranking, the level of agreement among the selection strategies improves. However, even at the top 40 threshold, the average index remains below 0.40, emphasizing substantial methodological differences in selection criteria. Phenotypic indices tend to prioritize progenies with superior agronomic traits under field conditions, while genomic approaches select based on predicted genetic values and underlying variance components. This divergence reinforces the importance of integrating both phenotypic and genomic information to improve the robustness of selection decisions in early-stage recurrent selection cycles.

These findings highlight the utility of coincidence analysis not only for assessing agreement among strategies but also for informing multi-criteria selection frameworks that balance short-term phenotypic gains with long-term genetic progress.

The predicted genetic gains indicate that the MMI and BI indexes were the most efficient in predicting genetic gains for both grain yield (GY) and ear weight (EW), with values surpassing those obtained with SHI. The superiority of the MMI index for GY (15.93%) and the close performance of BI (15.84%) suggest that these methods are more effective at combining multiple traits into a selection criterion that maximizes productivity. SHI, although traditionally robust, presented slightly lower gains (14.79%), reinforcing the notion that alternative indices can provide higher efficiency in specific contexts.

Inbreeding depression (ID) indicates a consistent reduction in deleterious additive gene effects and that the selected progenies are likely to suffer lower losses under inbreeding, a desirable outcome in pre-breeding programs. The ability to reduce inbreeding depression while improving grain yield highlights the potential of these selection strategies to simultaneously enhance performance and genetic stability. The inbreeding values results suggest that while these progenies from F_2_ commercial hybrids exhibit strong additive effects for grain yield, they may be more sensitive to inbreeding, a key consideration in early recurrent selection cycles.

The total predicted selection gain (PG total), reflecting the aggregate improvement across traits, was numerically greater with the Smith–Hazel index (28.25%), followed by the BI (26.37%) and MMI (26.27%), indicating that all three indexes were robust for multi-trait selection in half-sib progenies of tropical maize. Although SHI showed a numerically greater composite score, MMI and BI provided more favorable responses for key breeding objectives such as grain yield and reduction in inbreeding depression.

The findings of the present study align with those of Velmurugan et al. [13], who demonstrated significant genetic variability in BC_2_F_1_ maize progenies derived from *Zea nicaraguensis* crosses, particularly for kernel traits such as 100-seed weight, color, and shape. Our results and those of Velmrugan et al. [13] converge on the importance of maintaining and reintroducing genetic diversity as a strategy for improving maize performance under variable environmental conditions, and both highlight the value of multivariate analyses, such as mixed-model predictions, for guiding selection decisions in early recurrent selection cycles.

The REML/BLUP methodology was applied to estimate genetic gains and assess genotypic values among maize half-sib progenies under a recurrent selection scheme [5]. The REML/BLUP approach yielded high accuracy estimates and predicted substantial selection gains: 24.07% for EL, 21.88% for GY, and 18.23% for ED. These results underscore the efficiency of mixed models in quantifying genetic progress in maize breeding populations.

These findings reinforce the utility of BLUP-based methodologies in the estimation of genetic values and selection gains in maize, as implemented in our study. The positive association between yield components and GY through traits like ear length supports the integration of multivariate selection strategies, such as the SHI, BIA, and MMI indices, which incorporate multiple agronomic traits into the decision-making process. The genetic architecture observed by [5] aligns with our own findings of divergence among selection indices and suggests that trait-specific indirect selection paths may enhance the efficiency of breeding programs, particularly when the coincidence among selection methods is low.

The outcomes of our study, particularly the effectiveness of BLUP-based selection strategies in generating additive genetic gains and enhancing grain yield performance, are consistent with simulation-based insights presented by Vieira et al. [9]. Their review highlights that selection strategies incorporating genomic prediction tools—such as BLUP and RR-BLUP—are especially efficient when applied to traits with moderate to high heritability, as is typical for yield-related components in maize. In this study, the application of BLUP (PopBLUP_GY) resulted in superior grain yield compared to the open-pollinated control, validating simulation findings that emphasize the robustness of BLUP in handling polygenic traits across early recurrent selection cycles. This underscores the value of integrating both predictive models and cyclic selection schemes to simultaneously optimize short-term gains and long-term breeding potential in tropical maize.

The REML/BLUP framework has proven particularly effective in this context, providing precise estimates of additive genetic effects and enabling the identification of genotypes that combine high yield potential with resilience to inbreeding. As Kick and Washburn [18] demonstrated, predictive models that incorporate genotypic data can outperform phenotypic selection alone, particularly in unbalanced datasets or when dealing with complex traits.

In conclusion, the REML/BLUP-based evaluation provides a solid framework for informed selection decisions. Progenies with high additive genetic values and stable performance under inbreeding should be prioritized for hybrid development and population improvement, while lower-performing progenies may be strategically discarded or redirected to secondary breeding objectives.

The significance of the differences among new synthetic populations confirms the effectiveness of the selection and recombination processes applied to these new populations. Notably, PopMMI and PopBLUP_GY stood out as the top-performing groups, evidencing their potential for use in future hybrid combinations or for advancement as synthetic cultivars. These results reaffirm that the selection and recombination processes were effective in generating competitive and genetically diverse breeding materials.

As reported by Duarte and Sentelhas [20], the comparison with commercial hybrids is essential for determining the feasibility of replacing or complementing conventional materials with newly developed populations. In this case, although commercial hybrids still perform better in absolute terms, synthetic maize populations, especially those derived from the MMI and BLUP-based strategies, demonstrated yield levels that surpass traditional open-pollinated varieties and approach hybrid standards.

The genetic gains observed across progeny generations in our study are consistent with findings reported in a five-cycle intrapopulation recurrent selection program in half-sib progenies of fresh corn conducted in the southwest region of Goiás, Brazil. That study demonstrated the persistence of genetic variability and cumulative progress in key agronomic traits over multiple selection cycles, with heritability estimates exceeding 70% for most yield-related components, including ear diameter, marketable ear yield, and ear length [14]. Both studies underscore the efficacy of recurrent selection schemes in maintaining selection potential over time, even in populations derived from diverse genetic backgrounds, and highlight the utility of half-sib progeny structures for early-stage recurrent selection cycles and from F_2_ commercial hybrids. These parallels reinforce the strategic value of population improvement via intrapopulation recurrent selection, not only for specialty markets like fresh corn but also for broad-based grain yield improvement.

The observed results reinforce that the recombination of half-sib progenies, when guided by structured strategies such as REML/BLUP-based selection, leads to the formation of differentiated and agronomically competitive populations. Collectively, these findings support the continued use of the RV-02 germplasm base as a cornerstone for recurrent selection programs focused on improving tropical maize in terms of yield, resilience, and genetic progress across cycles and environments.

In summary, the data support the conclusion that genetic gains were successfully achieved, particularly in PopMMI and PopBLUP_GY, which surpassed or closely approached commercial check performance in yield-related traits. These results validate the effectiveness of selection strategies applied and demonstrate the potential of these populations for advancing synthetic variety development or serving as sources for inbred line extraction.

When comparing the relative positions of the synthetic populations based on predicted and realized grain yield (GY), clear differences emerge that highlight the limitations of the predictive models. PopMMI achieved the highest realized yield (6300 kg ha^−1^), although its predicted value (4590 kg ha^−1^) placed it close to PopBIA and PopSHI. This underestimation suggests that the index did not fully capture favorable non-additive effects or environmental contributions expressed in the field.

PopBLUP_GY presented the most striking shift, being ranked lowest in predicted yield (4025 kg ha^−1^) but outperforming PopBIA and PopSHI in realized yield (5620 kg ha^−1^). This discrepancy indicates that the BLUP-based model underestimated its genetic potential, likely due to unaccounted dominance and epistatic effects, which are not fully represented in additive genetic predictions, probably by early genetic cycles from F_2_ populations. Nevertheless, even though the predicted values were inferior to those obtained with phenotypic indices, the realized yield clearly demonstrates the efficiency of the BLUP-based method in selecting superior progenies with higher additive values. This outcome reinforces the strength of BLUP as a selection strategy, since its primary goal is not necessarily to maximize short-term prediction accuracy, but to identify and accumulate favorable additive effects that ensure sustained genetic progress across recurrent selection cycles.

Finally, the use of F_2_ populations derived from commercial hybrids to synthesize base populations for recurrent selection cycles in maize breeding can be considered a valuable strategy, as these populations harbor a high frequency of favorable alleles that have already been combined through intensive selection in commercial programs. This allows the synthesized populations to start from a high-performing genetic pool, thereby accelerating initial genetic gains. However, such populations may also display increased homozygosity and greater sensitivity to inbreeding, given that commercial hybrids are derived from specific combinations of inbred lines. In this context, the adoption of half-sib progeny selection schemes is technically advisable, since they reduce the rate of inbreeding compared with full-sib or inbred progenies, while maintaining sufficient genetic variability for future progress. Therefore, while F_2_ populations from hybrids can provide an efficient starting point for recurrent selection, their sustainable use requires strategies that balance the exploitation of favorable alleles with the preservation of long-term genetic diversity.

## 4. Materials and Methods

The study was carried out within a long-term maize breeding program in Southwest Goiás, Brazil, from 2016 to 2020 (Figure 4). In this region, the first season is sown in October of the previous year and harvested in February of the following year, while the second season is sown in February and harvested at the end of August. In 2016 (second season), the RV-02 population was planted in isolated plots to evaluate 182 half-sib families (HSP) across three locations, as described by Chavaglia (2016) [21]. In 2017 (second season), 32 superior HSP were recombined with 9 HSP previously identified for foliar disease resistance. In 2018 (first season), the recombination of 41 HSP generated a new population from which 194 HSP were selected. In 2018 (second season), 194 progenies were self-pollinated and obtained 194 S1 progenies. In 2019 (second season), these 194 HSP, together with two checks and their respective S1 progenies, were tested in field trials. In 2020 (first season), 38 selected HSP were recombined, resulting in 5 populations that were evaluated in 2020 (second season). These seven sown dates, with a chronological sequence of recombination, evaluation, and selection, ensured consistent replication across years, captured environmental variability, and established a robust framework for assessing the efficiency of the selection methods.

### 4.1. Recurrent Selection Cycle 0

The RV-02 Cycle 2 maize population was developed through a structured breeding and selection program over 15 years, serving as the genetic basis for the present study.

The original RV-02 Cycle 0 maize population originated from the work of Cárdenas [22], who collected germplasm from commercial maize fields in Southwest Goiás, Brazil, and established 14 populations (HG1 to HG14) from F_2_ commercial hybrids (Figure 5).

A full diallel (14 × 14) among 14 HG populations was conducted in January 2001 at the experimental field of the Department of Genetics, “Luiz de Queiroz” College of Agriculture (ESALQ/USP), in Piracicaba, São Paulo, Brazil (Figure 5). A total of 91 full-sib progenies were generated through manual crosses between adjacent population rows. Fifty to sixty ears per pair of rows were pollinated. Field trials with 14 HG populations and 91 full-sib progenies were conducted at Piracicaba (SP), Anhembi Farm (SP), and Rio Verde (GO), Brazil (Figure 5).

Two synthetic populations were formed: RV-01 from the recombination of HG-04, HG-05, HG-07, HG-10, HG-13, and RV-02 from HG-03, HG-06, HG-09, HG-11, HG-14 (Figure 5). The RV-02 synthetic population was selected for resistance to gray leaf spot (*Cercospora zeae-maydis*).

### 4.2. Recurrent Selection Cycle 1

The first cycle of recurrent selection of base population RV-02 (C1) was obtained under selection of 182 half-sib progenies (HSP) in three locations in the Southwest of Goiás, Brazil, in 2016, in the second season. These progenies were initially evaluated by Chavaglia [21] for grain yield and foliar diseases, including southern rust (*Puccinia polysora*), gray leaf spot (Pantoea ananatis), northern leaf blight (*Exserohilum turcicum*), cercospora leaf spot (*Cercospora zeae-maydis*), and maize bushy stunt (*Spiroplasma kunkelii*). In 2017, in the second season, 32 half-sib progenies were selected using the Mulamba and Mock [23] rank summation index, assigning a weight of 4 to grain yield and a weight of 1 to each disease variable. An additional 9 half-sib progenies were selected specifically for foliar disease resistance, resulting in 41 selected progenies. These progenies were recombined using the Irish method, in which a bulked pollen mixture from the progenies was used to pollinate each emasculated progeny. It originated from the Population RV-02 (Cycle 1).

In 2018, in the first season, 194 HSP were extracted from Population RV-02 (Cycle 1), and in the second season, the 194 HSP were self-pollinated and obtained 194 S1 progenies.

### 4.3. Recurrent Selection Cycle 2

The 194 HSP were evaluated in the 2019 second season using a 14 × 14 triple lattice design with two checks: a single-cross hybrid (DKB 290) and an open-pollinated variety AL Bandeirante. In parallel, the S1 progenies were grown in a separate field to evaluate inbreeding depression. To control competition effects, S1 rows were flanked by S0 base population rows, with a configuration ensuring uniform growth conditions.

The experiments with 194 HSP and 194 S1 progenies were conducted at the experimental field of the Federal University of Jataí, Jataí, Goiás, Brazil (17°53′ S, 51°43′ W, 780 m altitude, average annual precipitation ~1800 mm).

The plots consisted of a single 4-m row with 20 plants, spaced 0.90 m between rows and 0.20 m between plants. The following traits were evaluated: male flowering (MF), female flowering (FF), ear height (EH), plant height (PH), number of ears (NE), ear length (EL), ear diameter (ED), ear weight (EW), grain yield (GY), grain moisture (GM), and inbreeding depression (ID). The ID was calculated as:ID = 100 × ((m_0_ − m_1_)/m_0_), (1)
where 

m_0_ = mean of S_0_, and

m_1_ = mean of S_1_.

The second cycle of recurrent selection of Population RV-02 (C2) was obtained after recombination of 38 selected S1 progenies selected from 194 half-sib progenies in the 2020 first season (Figure 4). The 38 S1 selected progenies of each selection strategy in recurrent selection programs were recombined to generate five new populations: PopBIA (Base Index), PopSHI (Smith–Hazel Index), PopMMI (Mulamba–Mock Index), PopBLUP_GY (BLUP for grain yield), and PopBLUP_ID (BLUP for inbreeding depression).

### 4.4. Selection Strategies and Genetic Gain Prediction

Twenty percent of the best-performing progenies (38 out of 194 HSP) were selected based on five selection strategies: (i) phenotypic values using selection indexes Base Index, Smith and Hazel Index, and Rank Summation Index, and (ii) genotypic values (BLUPs) for grain yield and inbreeding depression. Selection indexes assigned weights of 2 (increase) for grain yield and 1 (decrease) for inbreeding depression. Genotypic selection considered these traits independently.

S1 progenies were used to calculate inbreeding depression and recombination, which were included as traits in all selection procedures. Selection indexes included the Base Index [24], the classical Smith [25] and Hazel [26] index (SHI), and the Mulamba and Mock [23] rank summation index (MMI):Base Index (BI): IBIj = ∑j x_j_ a_j_, (2)
where x_j_ are standardized trait means (per progeny) and a_j_ are predetermined economic/subject-matter weights.Smith–Hazel (SHI): ISHI = b′× I with b = P^−1^ Ga, (3)
where P and G are phenotypic and genetic (co)variance matrices, a is the vector of economic weights. I is the vector (t × 1) of values of genotype j.Mulamba & Mock rank-sum (MMI): IMMI = ∑j r_ij_ w_i_, (4)
where rij is the rank of genotype j for trait i, and wi denotes the weights. The lower ranks for inbreeding depression and higher ranks for grain yield were favored. Weights were set to 2 for grain yield and 1 for inbreeding depression (other traits’ weights = 0) in the phenotypic selection scenario. For MMI, lower ranks for inbreeding depression and higher ranks for grain yield were favored.

Genotypic evaluations used mixed models (REML/BLUP), with the following linear model:Y = Xr + Zg + Wp + Ti + Ɛ, (5)
where Y is the phenotypic vector;

Xr = random repetition effects;

Zg = random genotypic effects;

Wp = random plot effects;

Ti = fixed ancestry matrix; and

Ɛ = random residuals.

The fixed ancestry matrix coded the pedigree/background groups that formed the RV-02 population from F_2_ commercial-hybrid sources (HG03…HG14 as per the RV-02 synthesis described by Cárdenas [22]). These fixed effects were included to absorb systematic ancestry structure during REML/BLUP estimation.

Predicted gains were computed from index-based responses (BI/SHI/MMI) using selected-top-20% progenies and, for BLUP-based strategies, from the mean of selected BLUPs relative to the base mean, expressed as percent gain per trait; selection considered yield (weight = 2) and inbreeding depression (weight = 1) together in the index scenario, and separately in BLUP scenarios.

Realized gains were estimated after recombining the selected sets into five derived populations (PopBIA, PopSHI, PopMMI, PopBLUP_GY, PopBLUP_ID) and evaluating them in RCBD. The gains were computed as the difference (or % difference) between recombined population means and the reference/base.

The agreement among selection strategies over the universe of genotypes was quantified. For each strategy s and selection intensity k, the top-k selection set S_{s,k} was defined with cardinality k.S_{s,k} ⊆ U, |S_{s,k}| = k

Pairwise coincidence (overlap) between strategies p and q at intensity k was computed as:CI_{p,q}(k) = |S_{p,k} ∩ S_{q,k}|/k

When selected sets had different sizes, the normalized version was reported:CI_{p,q}(k) = |S_{p,k} ∩ S_{q,k}|/min{|S_{p,k}|, |S_{q,k}|}

As a sensitivity analysis, we also computed the Jaccard similarity:J_{p,q}(k) = |S_{p,k} ∩ S_{q,k}|/|S_{p,k} ∪ S_{q,k}|

To summarize stability across all M strategies, we used the mean pairwise coincidence:CI^¯^(k) = 2/[M(M − 1)] · Σ_{p < q} CI_{p,q}(k)

The top 10 until the top 40 corresponded to the highest-ranking subsets of progenies selected by each method. In practice, all progenies were first ranked according to their respective criteria, predicted genetic values for BLUP-based strategies or index scores for phenotypic indexes. From these ordered lists, the top 10 to the top 40 progenies were extracted, and pairwise coincidence indices were subsequently calculated to assess the degree of overlap among methods.

Estimations were performed using the lme4 package in R [27].

### 4.5. Recombination and Evaluation of Recombined Progenies

The five populations comprised the population RV-02 (C2), which was evaluated in 2020, in the second season. Field trials used a randomized complete block design (RCBD) with three replications and seven treatments. The plots with 4 lines, spaced by 0.9 m, were used and the same traits were measured in the HSP and S1 progenies of the RV-02 (C2) populations.

## 5. Conclusions

The evaluation of five selection strategies in maize recurrent selection showed that both phenotypic indexes (BIA, SHI, and MMI) and BLUP-based approaches (BLUP_GY and BLUP_ID) were effective in identifying superior progenies. Some progenies were consistently selected across all methods, confirming their reliability for recombination. Although predicted and realized values differed, particularly for BLUP_GY, the overall results indicate that phenotypic and BLUP-based strategies are complementary: phenotypic indexes provided higher predicted short-term gains, while BLUP approaches proved efficient in capturing additive effects and reducing inbreeding. Therefore, integrating both methods can strengthen recurrent selection by combining immediate realized gains with sustainable genetic progress for tropical maize improvement.

## Figures and Tables

**Figure 1 plants-14-03149-f001:**
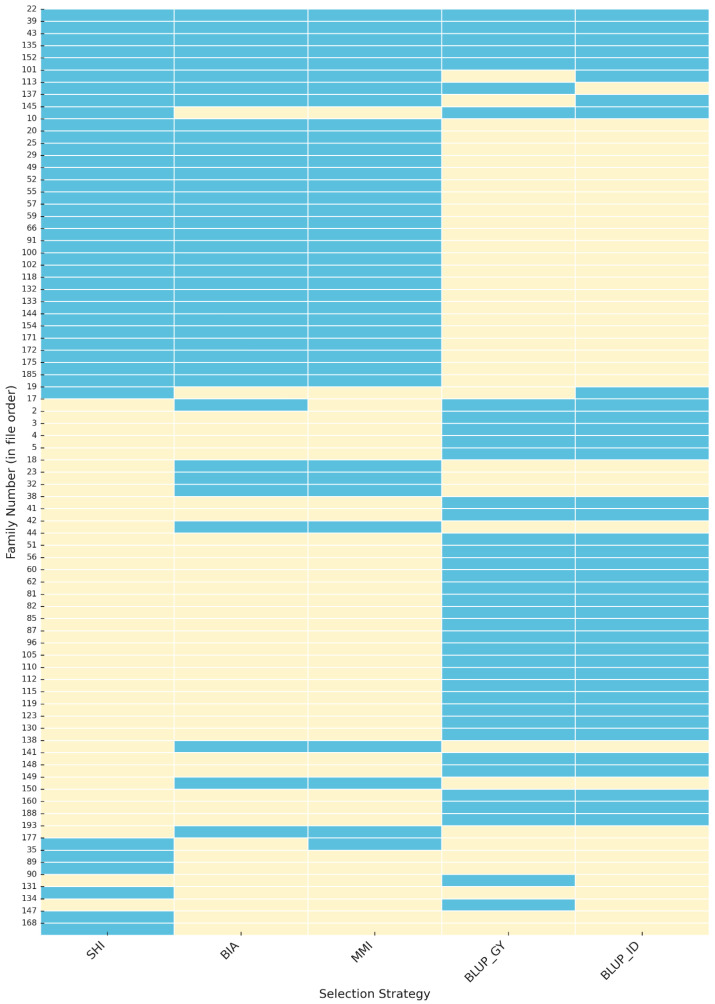
Heatmap of the maize progenies selected by five different breeding strategies: BIA (Base Index Approach), SHI (Smith–Hazel Index), MMI (Mulamba–Mock Index), BLUP_GY (REML/BLUP for Grain Yield), and BLUP_ID (REML/BLUP for Inbreeding Depression). Each cell indicates whether a specific line was selected by the corresponding method (blue = selected; yellow = not selected).

**Figure 2 plants-14-03149-f002:**
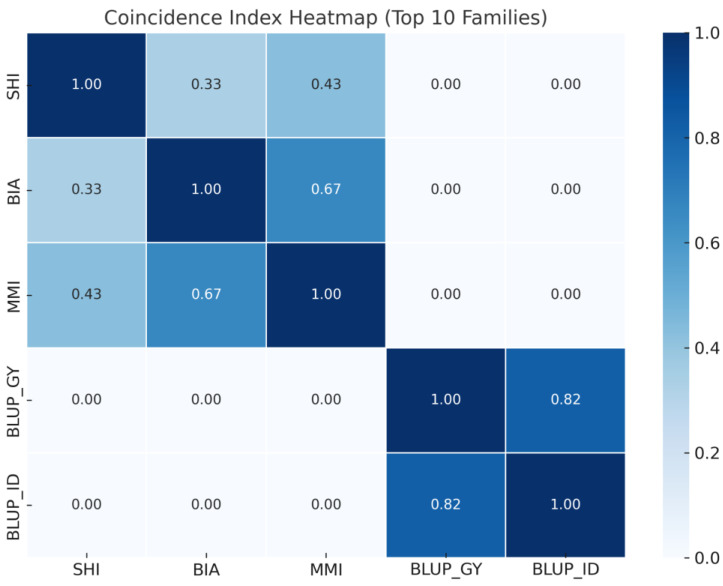
Heatmap shows the coincidence index among five selection methods—BIA (Base Index Approach), SHI (Smith–Hazel Index), MMI (Mulamba–Mock Index), BLUP_GY (REML/BLUP for Grain Yield), and BLUP_ID (REML/BLUP for Inbreeding Depression)—based on the top 10 most frequently selected maize progenies. Values represent the proportion of progenies jointly selected by each pair of methods.

**Figure 3 plants-14-03149-f003:**
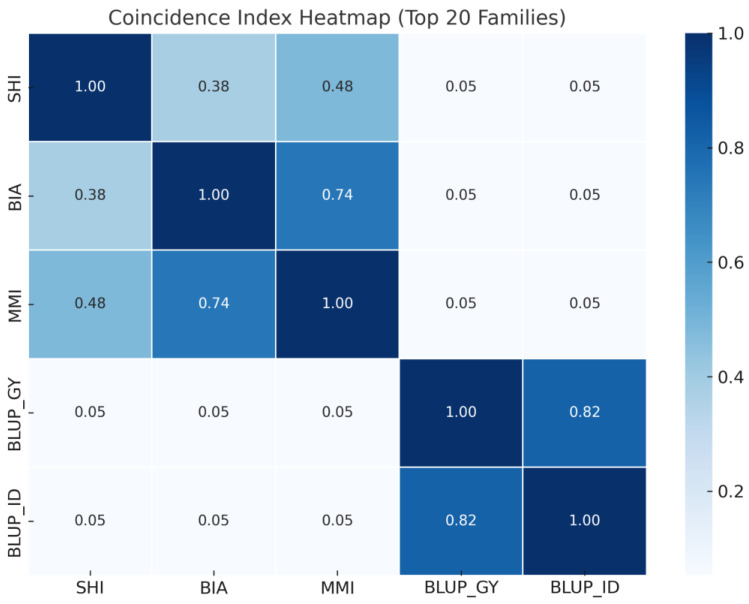
Heatmap presents the coincidence index among five different selection strategies for tropical maize based on the 20 most frequently selected progenies. It highlights the consistency or divergence in line selection among phenotypic indexes (BIA, SHI, MMI) and REML/BLUP-based methods (BLUP_GY and BLUP_ID). Higher values indicate greater agreement in line selection, supporting integrated use or comparative assessment of these approaches in breeding pipelines.

**Figure 4 plants-14-03149-f004:**
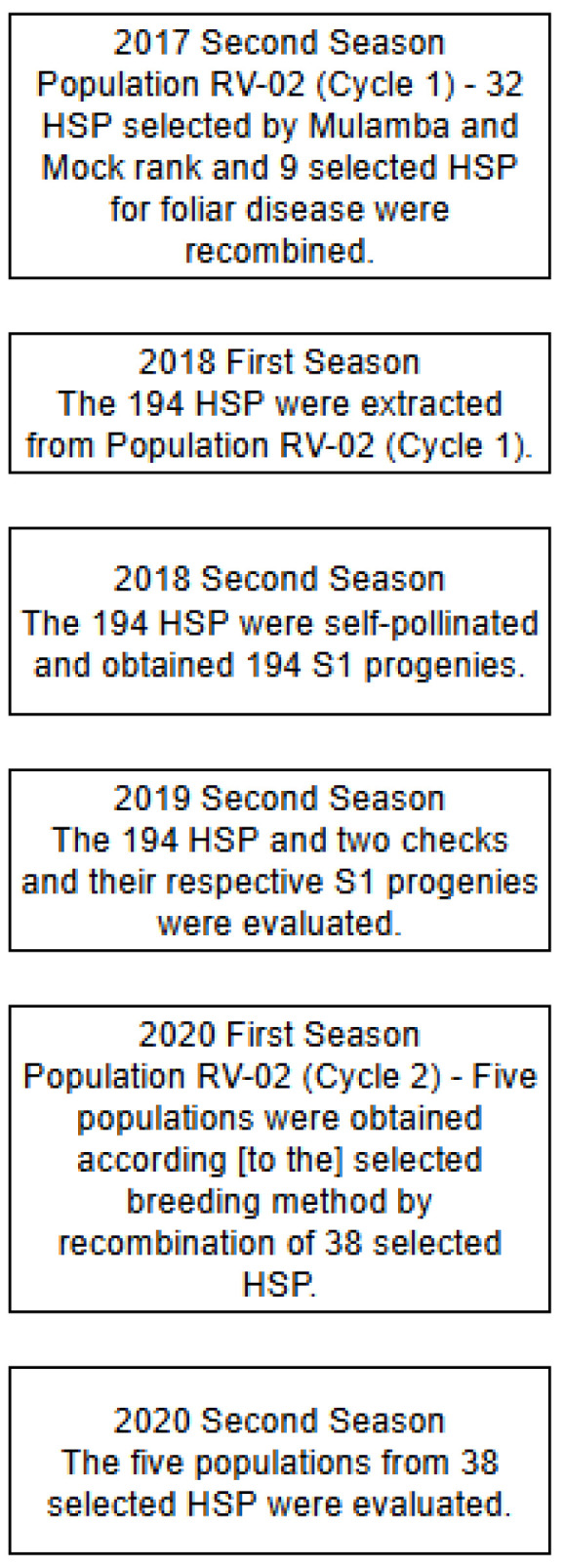
Flowchart illustrating the recurrent selection cycles of the RV-02 tropical maize population across seven sown dates.

**Figure 5 plants-14-03149-f005:**
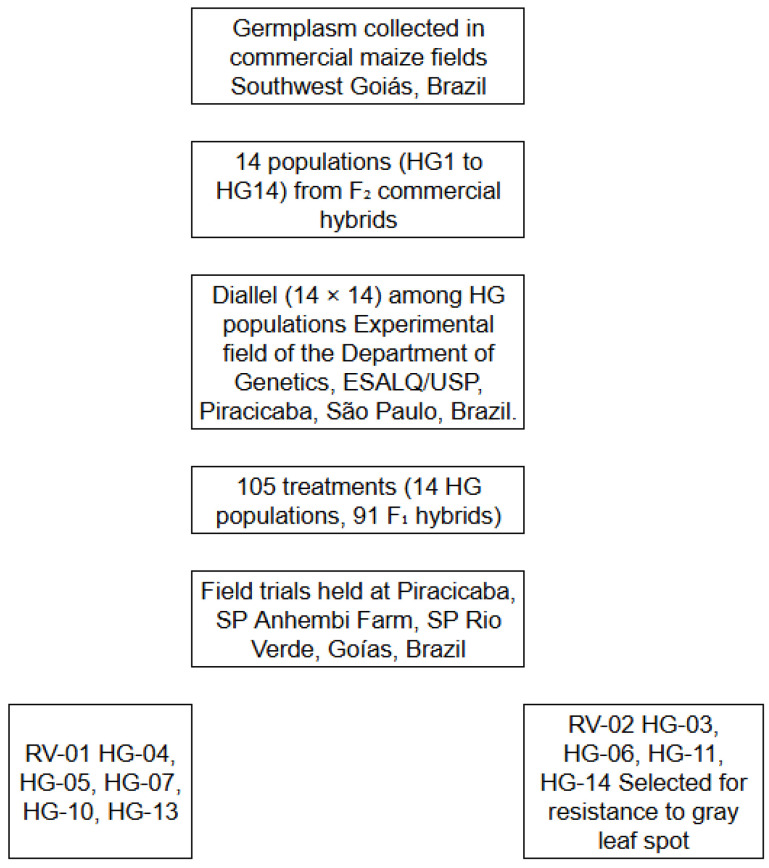
Flowchart of the initial development and selection process of RV-02 synthetic maize population.

**Table 1 plants-14-03149-t001:** Predicted selection gains (PG) of 38 selected progenies using the Smith–Hazel index (SHI), Base index (BI), and Mulamba–Mock Index (MMI) in 194 half-sib maize progenies.

Variable	SHI_Xs	SHI (%)	BI_Xs	BI (%)	MMI_Xs	MMI (%)
GY	4515	14.79	4597	15.84	4590	15.93
EW	5480	13.41	5534	14.01	5531	13.98
ID	36.44	−14.45	35.93	−14.96	35.82	−15.07
NE	14.92	7.56	14.62	6.19	14.61	6.11
ED	14.72	4.72	14.48	3.61	14.52	3.79
EL	4.40	2.17	4.37	1.75	4.36	1.71
MF	59.71	0.24	59.62	0.15	59.62	0.16
FF	58.50	−0.05	58.34	−0.19	58.42	−0.12
PH	2.250	−0.05	2.26	−0.05	2.25	−0.06
EH	1.010	−0.09	1.01	−0.07	1.01	−0.07

GY: grain yield (kg ha^−1^); EW: ear weight (kg ha^−1^); ID: inbreeding depression (% relative to the base population); NE: number of ears; ED: ear diameter (cm); EL: ear length (cm); MF: male flowering (days); FF: female flowering (days); PH: plant height (m); EH: ear height (m).

**Table 2 plants-14-03149-t002:** Predicted additive genotypic values (μ_i_ + g_i_) for grain yield and inbreeding depression (%) across three generations of main half-sib maize progenies for PopBLUP_GY, and BLUP_ID.

3rd G	Blup_GY	μ_i_ + g_i_	Blup_ID	u_i_ + g_i_
135	1423	4890	−17.3	33.7
145	1059	4525	−15.8	35.2
85	1020	4486	−10.9	40.1
150	746	4212	−11.2	39.8
87	706	4173	−10.6	40.4
4	626	4092	−9.4	41.7
44	604	4071	−9.1	41.9
82	580	4047	−8.6	42.4
96	568	4035	−8.5	42.5
105	565	4031	−8.4	42.6
Mean	558	4025	−7.9	43.1

**Table 3 plants-14-03149-t003:** Comparison of means for male flowering (MF), ear length (EL), ear weight (EW), and grain yield (GY) evaluated in five maize populations derived from different selection strategies (PopBLUP_GY, PopBIA, PopSHI, PopMMI, and PopBLUP_ID), along with two commercial cultivars (Hybrid P3898; open-pollinated variety AL Bandeirante).

Populations	MF Days	EL cm	EW (kg ha^−1^)	GY (kg ha^−1^)	Predict GY (kg ha^−1^)
P3898	67.83 a *	18.38 ab	11,620 a	9400 a	
PopMMI	66.67 ab	18.88 a	7880 b	6300 b	4590
PopBLUP_GY	66.33 ab	16.33 ab	7060 b	5620 b	4025
PopBIA	65.33 b	15.88 ab	5980 c	4760 c	4597
PopSHI	66.33 ab	17.17 ab	5900 c	4650 c	4515
PopBLUP_ID	66.33 ab	15.75 b	6100 c	4810 c	
AL Bandeirantes	66.67 ab	18.04 ab	6010 c	4480 c	

* Means followed by the same letter in a column do not differ significantly by Tukey’s test at 5% probability.

## Data Availability

The data presented in this study are available on request from the corresponding author.

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
