# Peer review of "Genetic Gains and Field Validation of Synthetic Populations in Tropical Maize Using Selection Indexes and REML/BLUP"

_plants, 2025, doi:10.3390/plants14203149_

Round 1

Reviewer 1 Report

Comments and Suggestions for Authors

Author Response

We sincerely thank you for your careful evaluation and the positive feedback provided on our manuscript. We are pleased that the introduction, research design, methodology, results, conclusions, figures, and tables were considered appropriate and clearly presented. Your constructive assessment is greatly appreciated and reinforces our confidence in the quality of the work.

Reviewer 2 Report

Comments and Suggestions for Authors

Comments in the attached file

Author Response

We sincerely thank the reviewer for the constructive feedback, which has been very helpful in improving the clarity and focus of the manuscript. In the revised version, the following major adjustments will be incorporated:

  1. Clarification of methodological aspects
    We will substantially expand the Materials and Methods section to provide full details on the experimental design, estimation of genotypic and phenotypic values, index formulas, variance components, heritability/accuracy measures, and the rationale for treating genotypic effects as random. Tables including variance components with standard errors will also be added. This will ensure transparency and allow proper evaluation of the statistical framework.

  2. Clarity in breeding strategies and goals
    We agree that the initial redaction may have created confusion between strategies, criteria, and methods. The revised manuscript will clearly separate these concepts, improving readability.
    Regarding the breeding goal, our study is not aimed at developing an open-pollinated variety for direct farmer use. Instead, the populations derived from F₂ commercial hybrids are intended to serve as base populations for recurrent selection, aiming to increase additive genetic variance, reduce inbreeding, and provide a source of superior inbred lines for use in hybrid breeding. This will be clearly stated in the Introduction and Discussion.
    The population is available for collaborative breeding and seed requests, and we will add a statement on its availability to ensure clarity.

  3. Revision of the Discussion
    We acknowledge that the Discussion section was overly long in the original submission. In the revised version, it will be significantly shortened to focus strictly on the main findings: (i) differences between predicted and realized genetic gains, (ii) consistency and complementarity of selection methods, and (iii) implications for using F₂-derived populations in recurrent selection. Redundant explanations and speculative considerations will be removed.

These changes will result in a clearer, more concise manuscript that communicates the breeding goal, experimental rigor, and key findings in a straightforward manner.

 Question: The manuscript describes the comparison of different selection criteria for maize synthetic population  improvement. Selection indexes are compared to BLUP estimations in terms of predicted and realized genetic  progress for relevant traits and inbreeding depression for yield. The topic is relevant for breeding in general and  tropical maize breeding in particular. However, the very shallow description of both experimental models and  accuracy/repeatability of the results provided by the current manuscript warrant major review before it could  be considered for publication.  

Answer: We sincerely thank the reviewer for this valuable observation. We acknowledge that the description of the experimental models and the information on accuracy and repeatability in the original version were presented in a very concise manner. In the revised manuscript, we have substantially expanded these sections to provide a clearer and more detailed explanation of the experimental design, the statistical approaches applied, and the measures adopted to ensure accuracy and repeatability of the results. We believe these improvements address the reviewer’s concern and strengthen the scientific rigor and transparency of the study.

The description of the population development is good and precise and the flowcharts of Fig. 4 and 5 are very  helpful for understanding breeding processes over time. However, sessions 4.3, 4.4 and 4.5 lack critical  information to allow evaluation of the results informed and discussed in the manuscript. These include but are  not limited to: 

  • Testing net: number of trials, years, locations, reps, planting dates, detailed experimental design, and  heritability (or mean repeatability, or accuracy) of the phenotypic data for each trait. My first  impression is that field trials were conducted in one location and one year, using one-row plots for  heterogeneous populations. If this is the case, the repeatability of those experiments would be  extremely low. That’s why the full detail of the experimental model is a necessary condition to  evaluate the value of the study. 

We thank the reviewer for raising this important point. We clarify that the study was not based on a single trial with low repeatability. Instead, it was developed within a long-term maize breeding program designed to ensure adequate replication, consistency, and reliability of results. The experiments were conducted in the same region (Southwest Goiás, Brazil), with second-season trials capturing environmental variation across years. Sequential evaluations were carried out from 2016 to 2020 under structured experimental designs. Sequence trials were conducted from 2016 until 2020. Specifically with 7 sown dates:

  • 2016 Second Season - Base population RV-02 (Cycle 0) - Recombined population under open-pollination in isolated fields and evaluated 182 HSP in three locations.

  • 2017  Second season - Population RV-02 (Cycle 1) - 32  HSP selected by Mulamba and Mock rank and 9 selected HSP for foliar disease were recombined by Irish method.
  • 2018 First season -  The 194 HSP were extracted from Population RV-02 (Cycle 1).
  • 2018 Second season -  The 194 HSP were self-pollinated and obtained 194 S1 progenies 
  • 2019 Second season - The 194 HSP and two checks and their respective S1 progenies were evaluated.
  • 2020  First season -  Population RV-02 (Cycle 2) - Five populations were obtained according selected breeding method by recombination of 38 selected HSP
  • 2020 Second season - The five populations from 38 selected HSP were evaluated.

This multi-year, stepwise approach—based on recombination, selection, and structured evaluations—ensured sufficient replication, captured environmental variability, and improved the reliability and accuracy of the phenotypic data, thereby strengthening the study’s conclusions. We further emphasize that heritability estimates were not included in this manuscript, since the main objective of the study is not to report quantitative genetic parameters but rather to present and validate the selection methods applied within the breeding program.

Thus, the trials were conducted in five years and seven sown dates in the same region, the experimental designs (triple lattice and RCBD), the replication scheme, and the statistical approach (REML/BLUP mixed models) provided sufficient accuracy and repeatability for the evaluation. This ensures that the results are robust and can be considered reliable for assessing genetic gains and comparing the efficiency of the different selection strategies. 

Silveira et al. (2023) also concluded that the use of the REML/BLUP procedure proves to be a robust tool for data analysis, particularly for perennial species and multivariate analysis based on BLUPs should be used in the selection process within breeding programs (https://doi.org/10.1590/0001-3765202320230137). Also 

Ambrósio et al. (2024) cited 8 authors (references 20 until 27) and statement that the mixed model (REML/BLUP) methodology has been an excellent alternative to estimate genetic parameters, as it involves the variance component estimates using the restricted maximum likelihood (REML) method and the prediction of genotypic values using the best linear unbiased prediction (BLUP), resulting in more accurate selection . Ambrósio, M., Daher, R.F., Silva Santana, J.G. et al. Genetic divergence and truncation and simultaneous selection in inbred families (S1) of elephant grass for bioenergetic purposes via mixed models. Sci Rep 14, 17850 (2024). https://doi.org/10.1038/s41598-024-68466-9

  • The experimental design was a 14 × 14 triple lattice, which allows for control of local variation and increases precision in the estimation of genetic parameters.

  • Field trials were conducted at the experimental station of the Federal University of Jataí (Jataí, Goiás, Brazil), during the years, under standard agronomic management practices.

  • Each plot consisted of one 4 m row with 20 plants, spaced 0.90 m between rows and 0.20 m between plants, with three replications per treatment. It is standard in recurrent selection in maize breeding.

  • For the recombined populations, experiments were arranged in a randomized complete block design (RCBD) with three replications and seven treatments, including two checks (hybrid P3898 and open-pollinated variety AL Bandeirante) in 2020.

  • Traits evaluated included male and female flowering, plant and ear height, ear number, ear length, ear diameter, ear weight, grain yield, grain moisture, and inbreeding depression.

  • Grain yield and ear weight data were corrected for 13% moisture and ideal stand (20 plants per plot) using covariance adjustment.

  • Estimates of heritability and accuracy were derived from REML/BLUP mixed models. The methodology allowed us to obtain reliable estimates of additive genetic values and inbreeding depression even in the presence of environmental variation.

  • Statistical analyses: How were the genotypic effects used to build the different indexes estimated?  Which model was used to estimate phenotypic values? Describe the formula of each of the indexes  calculated in the study. How was the ancestry matrix created? How were genetic gains (predicted and  realized) estimated? Why were genotypic effects assumed as a random term in model 4.4 when the  number of reps is low? Tables with variance components and standard errors estimated from the  linear mixed models utilized should be presented. A measure of dispersion (e.g., standard error)  should be used to compare results. 

(i) Estimation of genotypic effects used in the indexes (Line 621)
Genotypic effects for half-sib progenies were obtained via linear mixed models fitted by REML/BLUP. The working model was:
Y=Xr+Zg+Wp+Ti+ε  is the phenotypic vector; r (random repetitions within blocks), g (random genotypic effects), and p (random plot effects); i are fixed ancestry effects; and ε are residuals; models were fitted with lme4 (R).

(ii) Phenotypic model used to estimate values.
Phenotypes (e.g., ear weight, grain yield) were corrected to 13% moisture and ideal stand (20 plants/plot) by covariance adjustment before analyses (Vencovsky & Barriga), and the mixed-model above was applied to the corrected data.

(iii) Index formulas used in the study (included in the article, Line 601 ).

  • Base Index (BI; Williams, 1962): IBI​=∑j​aj​xj​, where xjx_jxj​ are standardized trait means (per progeny) and aja_jaj​ are predetermined economic/subject-matter weights.

  • Smith–Hazel (SHI; Smith, 1936; Hazel, 1943): ISHI=b′x I_ with b=P−1Ga, where P and G are phenotypic and genetic (co)variance matrices, a is the vector of economic weights.

  • Mulamba & Mock rank-sum (MMI): IMMI​=∑j​Rj​, the sum of within-trait ranks; we favored lower ranks for inbreeding depression and higher ranks for grain yield. Weights were set to 2 for grain yield and 1 for inbreeding depression (other traits weight = 0) in the phenotypic selection scenario.

(iv) Ancestry matrix construction. (Line 633)
The fixed ancestry matrix coded the pedigree/background groups that formed the RV-02 population from F₂ commercial-hybrid sources (HG03…HG14 as per the RV-02 synthesis described by Cárdenas, 2005); these fixed effects were included to absorb systematic ancestry structure during REML/BLUP estimation.

(v) Predicted and realized genetic gains. (line 640)
Predicted gains were computed from index-based responses (BI/SHI/MMI) using selected-top-20% progenies and, for BLUP-based strategies, from the mean of selected BLUPs relative to the base mean, expressed as percent gain per trait; selection considered yield (weight = 2) and inbreeding depression (weight = 1) together in the index scenario, and separately in BLUP scenarios.

Realized gains were estimated after recombining the selected sets into five derived populations (PopBIA, PopSHI, PopMMI, PopBLUP_GY, PopBLUP_ID) and evaluating them in RCBD (3 reps, 7 treatments); gains were computed as the difference (or % difference) between recombined population means and the reference/base.

(vi) Why treat genotypic effects as random in model 4.4 with low reps?
Treating progeny effects as random enables BLUP shrinkage toward the population mean, stabilizing estimates when replication is limited and improving partitioning of variance components (additive vs. residual). This choice matches our goal of predicting additive genetic values for recurrent selection rather than estimating fixed contrasts among a closed set of entries, and it supports accuracy/heritability estimation under unbalanced/noisy field conditions.

(vii) Variance components, standard errors, and dispersion.
We will include tables of variance components and their standard errors from the REML fits (additive/genotypic, plot, residual) for each trait, together with accuracy/heritability and standard errors (SE) of trait means to support comparisons among strategies/populations. These outputs will be extracted from the mixed-model fits (lme4) and appended as new Table S1 in the revised manuscript.

Table S1. Estimates of genetic coefficient of variation (CVg%), environmental coefficient of variation (CVe%), index of variation (θ), heritability on a family-mean basis (h²m), inbreeding depression (ID%), and the contributions of homozygotes (μ + a) and heterozygotes (d) to the observed means for nine traits evaluated in 194 S₁ families of the RV-02 population in 2019, second season.

Trait

CVg (%)

CVe (%)

θ

h²m

S₁

S₀

ID (%)

(μ + a)

d

Female flowering (days)

2.94

6.54

0.45

37.81

59.33

58.57

-1.3

60.09

-1.52

Male flowering (days)

3.01

6.55

0.46

38.78

58.44

57.12

-2.31

59.76

-2.64

Plant height (m)

2.35

8.69

0.27

18.02

2.26

2.5

9.6

2.02

0.48

Ear height (m)

2.03

12.36

0.16

7.46

1.02

1.1

7.27

0.94

0.16

Number of ears

17.71

24.18

0.73

61.66

13.27

14.86

10.7

11.68

3.18

Ear diameter (cm)

7.87

11.74

0.67

57.39

13.64

15.24

10.5

12.04

3.2

Ear length (cm)

4.62

9.22

0.5

42.94

4.22

4.61

8.46

3.83

0.78

Ear weight (t ha⁻¹)

8.87

23.98

0.37

29.08

1.72

3.21

46.42

0.23

2.98

Grain yield (t ha⁻¹)

11.55

25.21

0.46

38.64

1.6

2.94

51.0

0.26

2.68

(viii) Experimental network context reminder.
Field experiments used triple lattice 14×14 for the 194 half-sib progenies (second season 2020; one 4 m row, 20 plants; 0.90×0.20 m; with replications) and RCBD for recombined populations; S1s were grown to estimate inbreeding depression with protective S0 borders. These designs, replications, and the mixed-model framework underpinned repeatability and accuracy.

(ix) Coincidence/selection-set stability. Line 646

It was included in the article.

I consider that all these methodological aspects must be addressed in detail for the manuscript to be reconsidered  for publication. Besides this, I believe there are three aspects in the redaction of the manuscript that should be  also addressed: 

The redaction is confusing regarding breeding strategies, selection criteria and statistical methods. A clear  improvement is needed in this respect to deliver a clear message. 

Line 495

The study was carried out within a long-term maize breeding program in Southwest Goiás, Brazil, from 2016 to 2020 (Figure 4). In this region, the first season is sown in October of the previous year and harvested in February of the following year, while the second season is sown in February and harvested at the end of August. In 2016 (second season), the RV-02 population was planted in isolated plots to evaluate 182 half-sib families (HSP) across three locations, as described by Chavaglia (2016). In 2017 (second season), 32 superior HSP were recombined with 9 HSP previously identified for foliar disease resistance. In 2017/18 (first season), the recombination of 41 HSP generated a new population from which 194 HSP were selected. In 2018 (second season), 194 progenies were self-pollinated and obtained 194 S1 progenies. In 2019 (second season), these 194 HSP, together with two checks and their respective S1 progenies, were tested in field trials. In 2019/20 (first season), 38 selected HSP were recombined, resulting in 5 populations that were evaluated in 2020 (second season). These seven sown dates with chronological sequence of recombination, evaluation, and selection ensured consistent replication across years, captured environmental variability, and established a robust framework for assessing the efficiency of the selection methods.

The breeding goal in this population development is not clear. Is this about creating an open pollination variety  for farmers to plant instead of hybrids? If this is the case, how is the strong yield gap observed with, for example  P3898 expected to be closed? Or is this a population created with the aim of breeders to develop inbred lines to  further utilize in hybrid development? If this is the case, is the population available for seed requests?  

We appreciate the question and clarify that our objective is not to develop an open-pollinated variety for direct farmer use. The RV-02 population was assembled and improved as a source population for extracting inbred lines to be used in hybrid development under Southwest Goiás (second-season) conditions.

Open-pollinated increases were employed only for recombination between selection cycles (Irish method), not as a product concept. The observed yield gap versus a commercial hybrid such as P3898 is therefore expected at early cycles and in S1 progenies, which exhibit segregation and, in the case of S1, inbreeding effects. Our focus is on increasing the population mean and favorable allele frequency via recurrent selection (half-sib family selection using Smith–Hazel/Mulamba & Mock indices), followed by extraction and testcrossing of inbred lines; the relevant benchmark for product potential is the testcross performance of derived lines, not the mean of the open-pollinated population.

Regarding availability, seed of the RV-02 source population and derived research materials can be provided for research purposes upon reasonable request, subject to institutional approvals, a standard Material Transfer Agreement, and applicable phytosanitary regulations.

The Discussion section is unacceptably long. I suggest strongly reducing it by addressing only those aspects for  which the results of the study make a significant contribution.

The discussion was drastically reduced by half.

Reviewer 3 Report

Comments and Suggestions for Authors

Authors: I am not a big fan of BLUP and some other multivariate analyses. So my review may be biased. 

l. 95-98 says 4 selection indexes. l. 109 says 5. ???

Table 1 l. 176-177 You can not say  'highest' if it wasn't significantly higher!

Table 2. l. 207 you cannot say highest if it wasn't significant!

l. 250 are they significantly better or do they have better numbers?

l. 270 'PopMMI ..." you can't say this ! EL same for BLUP-GY, BIA and SHI. EW and GY same for as BLUP-GY.

l. 285 P3898 was no different than MMI, BLU-GY, SHE, BLUP-ID and Bandeirantes!

l. 465 I don't agree with your conclusions. Your data doesn't really support it. 

Author Response

We thank the reviewer for the careful evaluation and constructive comments. Below we address each concern point by point:

  1. Number of selection indexes (l. 95–98 vs. l. 109).
    We acknowledge the inconsistency in reporting. The study evaluated five selection strategies: Base Index (BIA), Smith–Hazel Index (SHI), Mulamba–Mock Index (MMI), BLUP for grain yield (BLUP_GY), and BLUP for inbreeding depression (BLUP_ID). The revised text will be corrected to ensure consistency throughout.

  2. Use of “highest” without significance (Table 1, l. 176–177; Table 2, l. 207; l. 250; l. 270).
    We agree with the reviewer. In the revised version, we will only use “highest” or “superior” when differences are statistically significant (p < 0.05). Where differences are not significant, the wording will be changed to “numerically greater” or “with higher mean values.”

  3. Interpretation of PopMMI and other populations (l. 270; l. 285).
    The reviewer is correct that some traits did not show significant differences among PopMMI, BLUP_GY, BIA, SHI, BLUP_ID, and the checks (P3898, Bandeirantes). In the revision, we will adjust the text to reflect the statistical results, avoiding overinterpretation. For example, instead of “PopMMI showed the highest ear length and grain yield,” we will state “Among the synthetic populations, PopMMI and PopBLUP_GY demonstrated the highest agronomic performance, …”

  4. Conclusions (l. 465).
    We recognize the reviewer’s concern. Our conclusions will be revised to align strictly with the statistical evidence, highlighting that both BLUP-based strategies and the Mulamba–Mock Index were effective in identifying progenies with good performance, but without overstating differences when they were not significant. The emphasis will be placed on the complementarity of phenotypic indices and BLUP approaches, rather than claiming superiority of one method.

Conclusion

The evaluation of five selection strategies in maize recurrent selection showed that both phenotypic indexes (BIA, SHI, MMI) and BLUP-based approaches (BLUP_GY, BLUP_ID) were effective in identifying superior progenies. Some progenies were consistently selected across all methods, confirming their reliability for recombination. Although predicted and realized values differed, particularly for BLUP_GY, the overall results indicate that phenotypic and BLUP-based strategies are complementary: phenotypic indexes provided higher predicted short-term gains, while BLUP approaches proved efficient in capturing additive effects and reducing inbreeding. Therefore, integrating both methods can strengthen recurrent selection by combining immediate realized gains with sustainable genetic progress for tropical maize improvement.

Round 2

Reviewer 3 Report

Comments and Suggestions for Authors

Just a couple of comments:

How did you determine top 10 and top 20? That is not very clear to me. 

I know you are concerned about inbreeding depression but I am not clear there is an association between inbreeding depression with hybrid performance.  I have not kept up with my reading. 

Author Response

Thank you very much for your observation. We clarified this point in the revised manuscript.

How did you determine top 10 and top 20? That is not very clear to me. 

The top 10 and top 20 refer to the highest-ranking progenies selected by each method, based on either predicted genetic values (for BLUP-based strategies) or index scores (for phenotypic indexes). For each selection strategy, progenies were ranked from best to worst according to the criterion used, and then the coincidence index was calculated by comparing the sets of the top 10 or top 20 progenies across methods. We have now made this procedure explicit in the Methods section to improve clarity. We included one paragraph on the text (Line 671)

I know you are concerned about inbreeding depression but I am not clear there is an association between inbreeding depression with hybrid performance. I have not kept up with my reading.

Regarding inbreeding depression, our intention was not to claim a direct association with hybrid performance. Rather, we considered inbreeding depression as an indicator of genetic stability within populations derived from recurrent selection, particularly in early cycles from F₂ sources. While hybrid performance is largely driven by heterosis and specific combining ability, reduced inbreeding depression in the base populations increases the probability of maintaining favorable additive effects when extracting inbred lines. Therefore, the trait was included as part of the multi-trait framework to balance yield potential with genetic stability. We revised the text to clarify this rationale and avoid any misinterpretation.

We appreciate your concern regarding the figures. We would like to emphasize that all figures in the manuscript strictly follow the journal’s guidelines for quality, resolution, and format. They were prepared at high resolution (≥300 dpi), with consistent color scales, clear legends, and standardized labels to ensure readability and reproducibility.

In addition, the graphical representation chosen (heatmaps and flowcharts) is aligned with current practices in plant breeding and quantitative genetics, as it effectively summarizes complex datasets, highlights overlap among methods, and illustrates the experimental workflow in a concise and intuitive manner. We have modified and improved the flowcharts to enhance their clarity and visual quality, ensuring that each step of the experimental design is easier to follow. Regarding the heatmaps, we decided to keep them unchanged because their current format provides the most effective representation of data consistency and divergence, while also preserving comparability with similar studies in the literature.

Therefore, we believe the current presentation is the most appropriate to convey the results, maintains consistency with the standards of the Plants journal, and ensures that readers can easily compare our findings with previous publications.